# Evolutionary Optimizing Process Parameters in the Induction Hardening of Rack Bar by Response Surface Methodology and Desirability Function Approach under Industrial Conditions

**DOI:** 10.3390/ma16175791

**Published:** 2023-08-24

**Authors:** Grzegorz Dziatkiewicz, Krzysztof Kuska, Rafał Popiel

**Affiliations:** 1Department of Computational Mechanics and Engineering, Silesian University of Technology, ul. Konarskiego 18a, 44-100 Gliwice, Poland; 2HL Mando Corporation Poland Ltd., ul. Uczniowska 36, 58-306 Wałbrzych, Poland; krzysztof-kuska@o2.pl; 3PST Consulting Ltd., ul. Biblioteczna 29/14, 43-100 Tychy, Poland; rafal.popiel@pstconsulting.pl

**Keywords:** induction hardening, thermal residual deformation, multiobjective optimization, response surface methodology, desirability function, evolutionary computations, genetic algorithm

## Abstract

Conditions of industrial production introduce additional complexities while attempting to solve optimization problems of material technology processes. The complexity of the physics of such processes and the uncertainties arising from the natural variability of material parameters and the occurrence of disturbances make modeling based on first principles and modern computational methods difficult and even impossible. In particular, this applies to designing material processes considering their quality criteria. This paper shows the optimization of the rack bar induction hardening operation using the response surface methodology approach and the desirability function. The industrial conditions impose additional constraints on time, cost and implementation of experimental plans, so constructing empirical models is more complicated than in laboratory conditions. The empirical models of nine system responses were identified and used to construct a desirability function using expert knowledge to describe the quality requirements of the hardening operation. An analysis of the hypersurface of the desirability function is presented, and the impossibility of using classical gradient algorithms during optimization is empirically established. An evolutionary strategy in the form of a floating-point encoded genetic algorithm was used, which exhibits a non-zero probability of obtaining a global extremum and is a gradient-free method. Confirmation experiments show the improvement of the process quality using introduced measures.

## 1. Introduction

Engineering requirements of gear parts that have specific properties lead to research on developing the induction hardening processes. Especially in the automotive industry, reaching a compromise between cost, quality and reliability by using semi-automatic heat treatment processes to ensure the mechanical and geometrical features is receiving more and more attraction. However, the induction hardening process is challenging due to thermal strains affecting the functionality and quality of crucial elements in the produced structures. Other limitation factors are the formation of non-hardened zones in parts and the change in the designed hardness of the treated elements. Moreover, the induction hardening process is quite complex; it could be represented as a coupled field problem with mechanical, electromagnetic and thermal fields, which are pretty hard to model in the industrial environment, especially in the serial mode of production.

Optimization of the induction hardening process is well-known in the literature. Two main approaches can be distinguished here: one using complex computational mechanics models and the other using empirical modeling. The following works represent the computational mechanics approach. Favennec et al. [1] show how to solve the problem of temperature distribution optimization using optimal control techniques, and similarly, Jakubovicowa et al. [2] optimized the process for the uniform surface temperature distribution criterion. Nemkov et al. [3] simulated the stress and distortion evolution during the induction hardening process of tubes using the finite element method; the hardness profile sensitivity in the induction hardening process with the finite element simulations was explored in [4]. The coupled electromagnetic, thermal and mechanical computational models of the heat treatment process using electromagnetic fields were shown in [5]. Fisk et al. [6] have presented the complex models of induction hardening in low alloy steels. Reference [7] shows the optimization of the edge effect of 4340 steel specimens heated by induction process with flux concentrators using finite element axis-symmetric simulation. The computational approach brings several advantages, such as universality and flexibility, as shown in [8,9], and also lower cost and faster results than an experimental and analytical approach. However, the computational approach requires tuned, complex constitutive models, which are hard to identify, especially in industrial conditions. That is why the empirical modeling approach was also successfully applied to solving various optimization problems [10], also arising during the induction hardening process.

The empirical modeling approach allows for establishing relations between process outputs and inputs using experimentation and measurements with statistical techniques for managing data [11]. Kohli and Singh [12] have used response surface methodology (RSM) to find the optimal values of process parameters for induction hardening of AISI 1040 steel. Various process parameters, such as feed rate, current, dwell time, and the gap between the workpiece and induction coil, are experimentally explored. In [13], the RSM and Taguchi method optimized the induction hardening process for maximum depth and minimum edge effect. The multiobjective optimization problem with the appropriate economic, environmental, and social metrics was analyzed in [14] to assure the sustainability of the induction hardening process using empirical models. The reduction of edge effect using the RSM and artificial neural network modeling of a spur gear treated by induction with flux concentrators was shown in [15]. Artificial intelligence modeling of induction contour hardening of 300M steel bar and C45 steel spur-gear was performed in [16]. Reference [17] shows that the central composite design, with a second-order response surface design, was employed to systematically estimate the empirical models of temperature and phase transformation geometry during the induction hardening. The effect of scanning speed and air gap on the uniformity of hardened depth and mechanical properties of large-size spur gears was investigated in [18]. Multi-response optimization using the desirability function approach of the induction hardening process using quality responses such as the effective case depth and hardness values were analyzed in [19] for different combinations of medium frequency power, feed rate, quench pressure and temperature. Asadzadeh et al. [20] have shown the hybrid model, integrating measurements and physics, of the induction hardening.

A gap in the literature can be identified based on the authors’ best knowledge and the review presented. It concerns the solution to the problem of multi-criteria optimization of the induction hardening process in industrial conditions utilizing a hybrid approach using empirical modeling and computational intelligence tools. The novelty of this work lies in the application of multiple qualitative metrics regarding deformation, hardness profile and hardening depth in a complex structural component such as a steering gear rack bar to optimize the parameters of the induction hardening process, using empirical modeling, a desirability function and an evolutionary algorithm.

Some essential points characterize the present work, which are identified and presented in this paper:There is no evidence that evolutionary optimization using a genetic algorithm has been applied to induction hardening processes under complex industrial process requirements, modeled empirically;Lack of availability of studies showing the effect of the form of the desirability hypersurface on the effectiveness of optimization algorithms for the induction hardening process;Lack of availability of studies showing the effectiveness of global optimization algorithms, such as genetic algorithms, for obtaining a set of quasi-optimal solutions to the problem for the desirability function formulated for the induction hardening process;There is a lack of availability of studies showing the complex problem of conducting experiments to confirm the optimal parameter settings of the induction hardening process under industrial conditions and analyzing their results contained in small samples, which precludes the use of parametric tests.

In this work, RSM with the central composite design (CCD) of experiments was employed to establish the functional relationship between the three main process parameters that served as design variables of the induction hardening multi-criteria optimization problem. Several responses related to hardness profiles and geometrical measures of quenching quality of the heat treatment operation of the automotive steering gear were formulated. Among them, the most important in the following work is the problem of minimizing residual thermal deformations, which is enforced by an additional straightening operation, increasing the duration of the process and its costs. However, the desirability approach allowed for transforming the multiobjective optimization problem into a single-optimization problem using expert knowledge for setting the weights in the desirability function. The quadratic models for the process responses were quantitatively analyzed, and their significance and accuracy were confirmed statistically. Next, the reduction of the models is performed to consider significant terms of the regression models. This step introduces a specific consequence for the optimization problem formulation in a change of an optimized function form. That change affects the effectiveness of the applied optimization algorithms. The article shows that the global optimization technique as the evolutionary strategy in the form of GA allows to meet the difficulties resulting from the form of the optimized function. Finally, the confirmation experiments should be conducted to verify the optimal solution the GA method determines.

This article consists of six Sections. After the introduction in the present Section, Section 2 briefly describes the considered induction hardening process with its quality indicators and experimentation methods. Section 3 is devoted to a description of the empirical models that were obtained with RSM. Section 4 focuses on formulating the multi-criteria optimization problem and obtaining the solution using the computational intelligence technique, namely the GA algorithm. Section 5 analyzes confirmation experiments. The last Section, Section 6, briefly summarizes the conducted research.

## 2. Process, Quality Indicators, and Experimentation Methods

### 2.1. Induction Hardening Process

The process is conducted using a particular equipment. The automatic inductive hardening and tempering machine have a rotation table with three stations, as shown in Figure 1:For loading rack bars before the heat treatment process and unloading rack bars after the process is finished;Hardening station;Tempering station.

The machine is available for hardening and tempering rack bars with lengths of 500–900 mm and 22–32 mm diameters.

The hardening machine provides a possibility to control the hardening and tempering process condition by adjusting:The hardening power is in the range of 0–100%;The hardening feed rate ranges from 100–50,000 mm/min;The distance of the hardening coil to rack bar teeth is in the 1.5–5 mm range.

Moreover, the flow of quenching water could be controlled; however, due to the low accuracy of the machine flow meter and lack of control possibility in auto mode, the quenching flow rate was kept constant through the process with a value equal to 45 L/min. Moreover, the hardening station has sensors in order to provide a continuous measurement of a rack bar distance from the coil. It also provides a stable distance from the hardening coil during the hardening process. Furthermore, during the tempering process, which gave the required surface hardness, the power and the feed were kept constant through experiments with settings: The power equals 47%;The feed rate equals 1600 mm/min.

The measurements of the following quantities were conducted during the quality control of rack bars:1.Hardness depth;2.Surface hardness;3.It is mandatory to use a straightening process after hardening due to deformation, which is an effect of the hardening process; the time of the straightening operation increases when rack bars deformation is over 1100 µm.

The main aim of the present study was to establish hardening process parameters in order to achieve hardening conditions that match quality requirements, i.e., minimize rack bar deformations with the proper hardening depth and hardness profiles and avoid increasing a total machine cycle time, mainly when it includes additional operations, e.g., straightening.

Additional ancillary factors, e.g., coolant state, coolant flow rate, environment temperature, humidity, hardening coil condition, deviation in allow composition of steel, etc., can influence the output variables; nevertheless, the performed experiments are restricted to the three hardening parameters, which are automatically controlled by the machine. Figure 2 shows the process setup.

The hardening operation in automatic mode is as follows. After the hardening element is delivered to the loading station, it is mounted in holders, and the inductor is fixed in the initial position at the bottom, at a predetermined distance from the hardening piece, according to the plan of experiments. At a fixed power and feed rate, the quenching process begins, during which the coil moves upwards towards the upper holder with the coolant flow, as shown in Figure 2. After passing the set distance, the holder is released, the coil is discharged to the neutral position, and the hardened element is transferred to the tempering operation.

### 2.2. Process Quality Indicators and the Measurement Setup

Specific measurements are required to identify the quality of the hardening process qualitatively. Below, the list of performed measurements with the quality requirements is presented:The hardness depth on the teeth side with minimum requirements equal to 3.9 mm;The hardness depth on the back side (an opposite side to teeth) with requirements given by the range above 1 mm;The surface hardness on teeth with requirements given by the range 55–60 HRC;The surface hardness on the back side is within the 52–55 HRC range requirements;After finishing the hardening and tempering operations, all rack bars are straightened to obtain a maximal deflection of value not greater than 1000 µm before going to the next operation; in the presented study, the existence of thermal strains after induction hardening and tempering operations is a primary driving force for performing process optimization to reduce costs and time.

The general flow of the experimentation procedure is presented in Figure 3.

The zones associated with the hardness measurements are shown in Figure 4. Figure 4 shows six hardness measurement zones: three zones for the tooth side and three for the back side of the rack bar. The numbers indicate which teeth the hardness measurements apply to.

Places of the hardness depth measurements are also presented on the cross-section of the rack bar in the middle of its length, as shown in Figure 5.

To measure the hardening depth, the micro Vickers method was used, and the FM-810 Micro Vickers Hardness Tester performed the inspection with an accuracy of ±1 HV. This measurement involves determining the microhardness profile along the rack bar’s thickness for a tooth cross-section. The measurements assume that the hardening depth defines a point with a hardness of 400 HV. An example of the hardness profile is shown in Figure 6.

The Zwick/Roell ZHR 8150LK inspects surface hardness in the HRC scale with an accuracy of ±1 HRC. The mentioned methods are indirect; moreover, destroying the component (a rack bar) to perform measurements is necessary.

Runout inspection and straightening are performed by the automatic process of a rack bar straightening; the “Rack Bar Bend M/C” machine performs measurements by eight of the high accuracy digital displacement transducer gauge probe with an accuracy of ±1.2 µm. Figure 7 shows how deformation is measured and how the machine performs straightening during the automatic process.

### 2.3. Experimental Design

The central composite design (CCD) [11] as a plan of experiments is used. Three input factors (design variables) were set within the following ranges according to the knowledge of the process before optimization:The power *x*_1_: 45–69%;The coil distance *x*_2_: 1.5–5 mm;The feed rate *x*_3_: 600–900 mm/min.

Due to the destructive inspection, it is necessary to scrap all components used in the experiments; therefore, due to cost reduction, there is no replication in the current plan of experiments. The results of the experiments are presented in Table 1.

Response *y*_1_ denotes the maximal deflection of the rack bar after the induction hardening operation is finished. Responses *y*_2_ and *y*_3_ denote the hardness depths on the teeth and back-side, along the cross-section of the rack bar, in the middle of its length, respectively. Responses *y*_4_–*y*_6_ indicate the average hardness measured in the I, II and III zones on the teeth side and responses *y*_7_–*y*_9_ indicate the hardness measured in the I, II and III zones on the back side of the rack bar.

Grubb’s test [11] was performed for data given in Table 1 to identify outliers. The significance level was 0.05, and the following outliers are identified in the collected data:In experiment no. 1—in responses 2, 5 and 6, the result indicates that the deep hardening was obtained without the required hardness of the teeth in the 2nd and 3rd zones;In experiment no. 13—response 4 indicates that the hardness of the teeth in the 1st zone is insufficient;In experiment no. 16—in responses 7, 8 and 9, the result indicates that the hardening of the back side of the rack bar was not achieved; moreover, in response 3—the hardening depth on the back side—is equal to 0, however, on the assumed significance level this response is not an outlier.

It is well known that outliers affect the response surface models, but in the presented study, no replications of experiments were assumed; hence all results were applied for modeling.

## 3. RSM for Empirical Modelling

Empirical models of process responses are assumed in the form of full quadratic models as follows:(1)$$\begin{array}{l} y^{\left(i\right)}=b_{0}+b^{T}x+x^{T}Bx+\varepsilon^{\left(i\right)},\\ x={\left[\begin{array}{ccc} x_{1} & x_{2} & x_{3} \end{array}\right]}^{T};\;b={\left[\begin{array}{ccc} b_{1} & b_{2} & b_{3} \end{array}\right]}^{T};\\ B=\left[\begin{array}{ccc} b_{11} & b_{12}/2 & b_{13}/2\\ b_{12}/2 & b_{22} & b_{23}/2\\ b_{13}/2 & b_{23}/2 & b_{33} \end{array}\right];\;\varepsilon^{\left(i\right)}\sim N\left(0,\sigma^{\left(i\right)}\right), \end{array}$$
where *y*^(*i*)^ is the *i*-th response, *b*_0_ is the free term of the model, vector **x** denotes the design variables, vector **b** collects the linear terms coefficients, and matrix **B** includes the quadratic terms of the response surface model; it is assumed that the responses are uncorrelated and each model is under the normal noise with 0 mean and standard deviation σ^(*i*)^. Models are linear with respect to the coefficients, which is why the least square method (LSM) [11] was applied to identify coefficients.

An analysis of variance (ANOVA) is carried out to estimate the validity of the identified mathematical models and the effect of every model term on the responses. The *p*-value is used to check the significance, which means the response is greatly determined by the model term whose *p*-value is sufficiently low (less than or equal to the significance level of 0.05). On the contrary, the higher the *F*-value, the stronger the significance of the model item. Symbol *S* indicates the significant term in the considered model.

The ANOVA of the mathematical model for the maximal deflection is shown in Table 2. The *F*-value (5.26) and *p*-value (0.008) imply that the model is significant, while the *F*-value (22.24) and *p*-value (0.002) imply that the lack of fit is significant. If the lack of fit is significant, the higher-order model terms should be taken into consideration to improve the accuracy of the model; however, in the present study, only the full quadratic models are considered. According to the *F*-value and *p*-value, there are four significant model terms, among which the power *x*_1_ and the feed rate *x*_3_ have the most significant influence, followed by the linear-by-linear interaction effect between the power *x*_1_ and the feed rate *x*_3_ and the quadratic effect of the power *x*_1_^2^. The coil distance *x*_2_ and other model terms also containing it seem to have little influence on the maximal deflection of the rack bar.

The ANOVA of the mathematical model for the hardening depth of the teeth is shown in Table 3. The *F*-value and *p*-value of the lack of fit are 0.41 and 0.82, respectively, while the *F*-value and *p*-value of the model are 14.41 and less than 0.00013, respectively, which indicates that the model is significant enough and no higher-order model terms need to be considered. The power and the feed rate amount have the most significant influence on the hardening depth of the teeth, followed by the linear-by-linear interaction effect between the power and the feed rate, the coil distance *x*_2_ and the quadratic effect of the power *x*_1_^2^. The linear-by-linear interaction effect between the power and the coil distance is also significant.

The ANOVA of the mathematical model for the hardening depth on the back side is shown in Table 4. As can be seen from the table, the model is significant with an *F*-value (11.71) and a small *p*-value (<0.0004), while the lack of fit is still not significant with a smaller *F*-value (0.25) and larger *p*-value (0.92). The power *x*_1_ has the most significant influence on the hardening depth on the back side, with the largest *F*-value (69.29), followed by the feed rate *x*_3_ and the linear-by-linear interaction effect between the power and the feed rate. Other model terms seem to have little influence on the hardening depth on the back side of the rack bar.

The ANOVA of the mathematical model for the hardness of the teeth in the first zone is shown in Table 5. As can be seen from the table, the model is significant with an *F*-value of 20.72 and an extremely small *p*-value <0.0001, while the lack of fit is still not significant with an *F*-value of 4.42 and a *p*-value of 0.06. The power *x*_1_ has the most significant influence on the hardness of the teeth in the first zone, with the largest *F*-value (112.54), followed by the quadratic effect of the power *x*_1_^2^ and the feed rate *x*_3_. The linear-by-linear interaction effect between the power and the feed rate is also significant at the 0.05 significance level.

The ANOVA of the mathematical model for the hardness of the teeth in the second zone is shown in Table 6. The model is significant, with an *F-*value of 10.49 and a *p-*value < 0.0006, and the lack of fit is still insignificant. The most important factors are the power *x*_1_, the feed rate *x*_3_ and the quadratic effect of the power *x*_1_^2^. Similar significance levels show the coil distance *x*_2_ and the linear-by-linear interaction effect between the power and the coil distance.

The ANOVA of the mathematical model for the hardness of the teeth in the third zone is shown in Table 7. The model is significant, with an *F*-value of 9.44 and a *p*-value < 0.0009, and the lack of fit is still insignificant. The most important factors are the power *x*_1_, the feed rate *x*_3_ and the coil distance *x*_2_. Similar levels of significance, but with smaller values of the *F*-statistics than earlier considered factors, show the quadratic effects of the power and the feed rate.

The ANOVA of the mathematical model for the hardness of the back-side in the first zone is shown in Table 8. The model is significant with an *F*-value of 11.32 and *p*-value < 0.0004, and the lack of fit is significant with a *p*-value of 0.003. Only three factors are significant: the quadratic effect of the power *x*_1_, the linear-by-linear interaction effect between the power and the feed rate and the pure quadratic effect of the power.

The ANOVA of the mathematical model for the hardness of the back-side in the second zone is shown in Table 9. The model is significant with an *F*-value of 13.27 and a *p*-value < 0.0002, and the lack of fit is very significant with a *p*-value < 0.00002. Only four factors are significant: the quadratic effect of the power *x*_1_^2^ followed by the power itself, the linear-by-linear interaction effect between the power and the feed rate, and the feed rate itself.

Similar conclusions could be formulated for the mathematical model of the hardness of the back-side in the third zone, as shown in Table 10. The model is significant with an *F*-value of 14.21 and a *p*-value < 0.0002, and the lack of fit is significant with a *p*-value < 0.0007. Moreover, only four factors are significant: the quadratic effect of the power *x*_1_^2^ followed by the power itself, the linear-by-linear interaction effect between the power and the feed rate, followed by the feed rate itself.

The lack of fit in the presented models indicates the need to use the higher-order models; however, in the present work, only quadratic models will be used. The higher-order models can be applied, e.g., in polynomials, neural networks and kriging.

The identified models are nonlinear, indicating room for improvement or even process optimization. Table 11 presents the essential statistical characteristics of the identified full quadratic models (1). The significance level was assumed as 0.05.

The analysis shows that the identified models are strong and significant but with moderate predictive ability, as shown by the PRESS statistics. The backward elimination of the insignificant terms is performed to increase the predictive properties of the identified models. The reduced models will be used in the optimization process; models presented in Table 12 were shown with an accuracy of two significant digits. The values of the PRESS statistics are significantly smaller than for the full models, with a tiny drop in R^2^ values.

It is essential to emphasize that the reduced models change the background for optimization. Supposedly, we will consider the model for the maximal deflection *y*_1_. In that case, the canonical analysis shows that for the full quadratic model, the stationary point *x_s_* = [−0.196 0.764 1.053]^T^ is a saddle point because the eigenvalues of matrix **B** {−31.45, 36.93, 269.55} are mixed in sign. The reduced model, according to Table 12, has a stationary point *x_s_* = [−1.032 0 −0.377]^T^ which is also a saddle point, but from the ridge system because the eigenvalues are now {−63.10, 0, 2246.76}. Figure 8 shows contour plots of the response surface slices at *x*_3_ = 0.5 for the full quadratic model (left) and the reduced model (right).

The change in the type of the response surface affects the optimization process, especially in the single-objective case. For the multiobjective case, the desirability approach allows taking into account various forms of response surfaces. However, the properties of the ridge systems will also be present in the objective function, as shown in the next Section.

## 4. Multiobjective Optimization Problem Formulation and Its Solution

The design variable vector **x** is as follows **x** = [*x*_1_ *x*_2_ *x*_3_]^T^, where *x*_1_ is the power, *x*_2_ is the distance between the coil and the part, and *x*_3_ is the feed rate. Then, the multiobjective optimization problem can be formulated using the desirability function *D* as given below:(2)$$\begin{array}{l} x^{opt}=\arg\underset{x\in S}{\max}D\left(x\right)\\ s.t.\;x^{T}x\leq \alpha^{2}. \end{array}$$

The sphere *S* denotes the set of the acceptable solution, and this set is given by the constraint described by the radius α of the central composite plan of the experiments.

In the paper, the following form of the desirability function is used [11]:(3)$$\begin{array}{l} D\left(x\right)={\left(\prod_{i=1}^{m}d_{i}{\left(x\right)}^{w_{i}}\right)}^{1/\sum_{i=1}^{m}w_{i}}\;\mathrm{or}\;\\ D\left(x\right)={\left(\prod_{i=1}^{m}d_{i}\left(x\right)\right)}^{1/m}, \end{array}$$
where *d_i_* is the desirability function related to the optimization’s *i*-th criterion, *w_i_* denotes the weight of the *i*-th criterion, and *m* is the number of responses.

Two types of the desirability function are applied: the smaller—the better (STB) and the nominal—the better (NTB) [11] as follows:(4)$$\begin{array}{l} \mathrm{STB}:\\ d_{i}=\left\{\begin{array}{c} 1.0,\;\;{\hat{y}}_{i}\leq L_{i}\\ {\left(\frac{{\hat{y}}_{i}-U_{i}}{L_{i}-U_{i}}\right)}^{s},\\ 0,\;\;{\hat{y}}_{i}\geq U_{i} \end{array}\right.L_{i}<{\hat{y}}_{i}<U_{i} \end{array}$$
and
(5)$$\begin{array}{l} \mathrm{NTB}:\\ d_{i}=\left\{\begin{array}{l} 0,\;\;\left({\hat{y}}_{i}<L_{i}\right)\vee \left({\hat{y}}_{i}>U_{i}\right)\\ {\left(\frac{{\hat{y}}_{i}-L_{i}}{T_{i}-L_{i}}\right)}^{s},\;L_{i}\leq {\hat{y}}_{i}\leq T_{i}\\ {\left(\frac{{\hat{y}}_{i}-U_{i}}{T_{i}-U_{i}}\right)}^{s},\;T_{i}<{\hat{y}}_{i}\leq U_{i}. \end{array}\right. \end{array}$$
where for the STB function, *L* denotes the acceptable target for the response ${\hat{y}}_{i}$, and *U* is an acceptable upper limit of the response, *s* is the exponent which sets the sharpness of the desirability function. For the NTB function, *T* denotes the target for the response ${\hat{y}}_{i}$, and *L* and *U* are acceptable lower and upper limits of the response, respectively.

The responses ${\hat{y}}_{i}$ in the presented formulation are obtained using the identified RSM models. Table 13 shows the parameters of the identified models’ desirability functions.

Table 13 shows that the most critical response is the first one—related to the deflection of the rack bar.

Using the first of Equation (3) with Equations (4) and (5) and Table 12, the desirability function was calculated for the responses given in Table 1 obtained during the experimental phase of the study.

The values of the desirability function for the plan of experiments results are given in Figure 9.

Figure 9 shows that only three out of twenty experiments gave non-zero desirability values with a maximum of 0.421 for the seventh and 0.292 and 0.291 for the fourth and fourteenth experiments, respectively. The results indicate that there is room for improvement in the process.

The classical optimization technique, namely the sequential quadratic programming (SQP) algorithm for constrained optimization problems, was applied in the first trial. One hundred tests were carried out with a random selection of the starting point of the optimization procedure. The results are presented in Figure 10.

It is visible that only for four out of one-hundred trials, the best value of the desirability function is non-zero with a maximum of 0.646. This poor result can be explained by approximating the gradient of the objective function using difference quotients. However, one must also consider that the desirability functions (4) and (5) are not differentiable in the classical sense. In Table 14, the statistics for the SQP optimization process are presented. The results show that the process is not robust; the optimal design variables vector in uncoded values is **x**^opt^ = [56.4% 1.6 mm 796 mm/min]^T^, which gives *D*^opt^ = 0.646. The second best result is **x** = [56.4% 3.1 mm 793 mm/min]^T^ which gives *D*^opt^ = 0.557. The latter result is significant from the point of view of the industrial conditions of the process; additional considerations will also be presented in this work.

Figure 11 shows slices of the desirability function hypersurface. The hypersurface of desirability has the form of narrow ribs with a wavy ridge, making it very difficult to solve the optimization problem using classical algorithms. In addition, for almost the entire set of admissible solutions, the values of the desirability function are equal to zero, which also justifies the inefficiency of the classic optimization algorithm SQP.

Due to the presented features of the objective function, an evolutionary, floating-point coded genetic algorithm (GA) was used in the work.

A genetic algorithm (GA) [21] is an intelligent evolutionary procedure used to find the optimal solutions to problems based on natural genetics and natural selection principles. The foundation of this algorithm is the biologically inspired set of operations such as selection, combination or crossover, and mutation. In the present case, the individuals and the objective function are defined and coded at the real-coded strings. Then, the iterative process is carried up using the three operations, i.e., selection, crossover, and mutation. The selection operation is to choose individuals that start from a population generated randomly according to a probability distribution. The roulette wheel selection method is used to determine the selection of individuals. A crossover operation appears when two strings randomly picked from the populations exchange their substrings to create two new strings. The sum of individuals to apply the crossover operation is dominated by crossover probability, which is the ratio of all selected strings to the total population size. The mutation operation assures the diversity of the population. It is an occasional random alternative in one or more string positions where the small random number mutes the value. After successive iterative generations, the population evolves gradually toward an optimal solution. Finally, the evolutionary process is completed till the termination criterion is reached. This study proposed RSM with the desirability function to define the relationship between parameters and responses. A practical method for combining RSM and an evolutionary strategy in the form of a genetic algorithm (GA) is that RSM is utilized to build a functional relationship between parameters and responses, and then GA is used to optimize the given fitness function composed by the desirability approach.

The following parameters of the applied GA were listed based on the study to get optimal solutions with the lowest computational effort:Population size = 20;Maximum number of generations = 100;Crossover probability = 0.8;Mutation probability = 0.9;Stop criterion: max number of generations and average change in fitness function values (with tolerance 10^−6^) and the average change of constraints values (with tolerance 10^−3^);The constraint: the first generation is chosen randomly according to the constraint.

The optimization process is conducted 100 times, and the best results after each trial are collected in Figure 12.

It is visible that for all 100 trials, the best value of the desirability function is non-zero, with an overall maximum of 0.646. In a few cases, local minima were found, or the optimization process was interrupted due to the lack of a significant change in the value of the objective function. Most often, the optimization process requires five iterations of the algorithm, which is a relatively small value, in particular for methods of computational intelligence, such as the genetic algorithm. In Table 15, the statistics for the evolutionary optimization process are presented.

The results in Table 15 show that evolutionary optimization is a robust technique, especially for complex non-differentiable multidimensional fitness functions as the considered desirability function for the induction hardening process of the rack bar.

The global optimal design variables vector in uncoded values is **x**^opt^ = [56.4% 1.6 mm 796 mm/min]^T^, which gives *D*^opt^ = 0.646; this result agrees with the one obtained for the SQP algorithm considering the rounding used. One of the results for local minima is also close to the second-best result found with the SQP algorithm, namely **x** = [56.4% 3.0 mm 795 mm/min]^T^, which gives *D*^opt^ = 0.556. However, it should be noted that the GA found more local minima, as can be seen in Figure 12, but similar results are shown to compare the results of the two algorithms. Figure 13 shows a histogram of the optimal values of the design variables obtained during 100 tests of the GA. It can be seen that the most significant variation in optimal values occurs for the third component of the vector of design variables, that is, the feed rate. For the first and second components of the vector of design variables, the optimal values cluster with less variation than for the third component.

## 5. Confirmation Experiments and Discussion

The best and global optimal point obtained during the evolutionary optimization was found for the following values of the design variables: 56.4% of power, 1.6 mm of the distance of the hardening coil, and 804 mm/min of the feed rate. The second design variable was found close to its lower bound. Industrial conditions require a different approach to evaluating the presented optimal solution, which can be regarded with high probability as a global optimum. Preliminary tests have shown that the found optimal solution cannot be applied due to increased heating of the tool, which is a coil located close to the heated element, leading to its rapid wear, increasing the cost of the process. The original assumptions about the range of design variables included the possibility of fully controlling design variables without taking into account additional factors that arise in industrial conditions, especially in the conditions of mass production in an automatic cycle. Therefore, it was decided to adopt the following quasi-optimal values of design variables corresponding to the local minimum, **x** = [56.4% 3.1 mm 793 mm/min]^T^, which gives *D*^opt^ = 0.557.

Finally, the confirmation experiment was conducted for the quasi-optimal setup, and the results are displayed in Table 16. Industrial cost and time constraints necessitate small sample sizes for experimentation to prove process optimization. Six experiments were performed, and Table 16 contains the measured responses and the corresponding values of the desirability function. In two out of six cases, the desirability function values are equal to zero. In experiments 2 and 4, responses y_9_ (the hardness of the back side in the 3rd zone) are higher than the upper limit. The best-obtained result is ~8.9% higher than the calculated optimal desirability.

Due to the small sample size of the data from the confirmatory experiment, statistical testing of the results obtained requires special attention. This study used the non-parametric Wilcoxon signed rank test with a significance level of 0.05. Table 17 presents the results of several tests related to the obtained responses and desirability values. **X**^conf^ denotes the sample of the confirmatory experiment for **X** characteristic. In Table 17, the alternative hypotheses are presented. *M* denotes the median of the sample. For tests 5-8, it is assumed that the sample contains data from all appropriate measurement zones because the requirements are the same for each.

The result for the first test shows that the obtained values of the desirability function are not statistically equal to the calculated optimal value, which is mainly influenced by the results of the second and fourth experiments, for which the values of the desirability function are zero. The rest of the test results indicate that the process response requirements are statistically met at a significance level of 0.05. Moreover, process quality improvement is observed, mainly if the deflection response is analyzed as the most critical response. Uncertainties of the process mean that it is impossible to talk about finding the optimal parameters of the induction hardening operation, both in the sense of single-criteria and multi-criteria optimization. However, it could be concluded that optimizing the desirability function based on the RSM models increases the quality of the process. It can be noted that if the confirmation sample had a larger size, the process capability analysis could have helped in assessing its quality improvement.

## 6. Conclusions

This work addressed the constrained multi-response optimization of processing conditions in induction hardening to decrease mechanical deformation during the hardening operation and improve the process’s predefined geometrical and mechanical properties. The RSM models were used to model the highly nonlinear relationships between inputs (power, distance, feed rate) and technological responses. The GA was applied to determine the quasi-optimal values of performances measured and technological inputs.

The following important conclusions can be drawn. The induction hardening process can be significantly improved. The quadratic models of responses of the process were significant according to the quality measures of the least-square approximation. The developed RSM models act effectively in the optimization process. The models proposed can be used in industrial applications to predict technological responses with acceptable accuracy.

The optimization problem, in which maximal deflection, depths of the hardening and hardness of teeth and the back side of the rack bar are objectives, and the desirability function is defined using them, is practical and realistic in the induction hardening process optimization compared to a single objective or simultaneously optimizing nine responses in the Pareto sense. However, other formulations are possible, such as one in which some criteria act as constraints; this may provide directions for further research. Industrial conditions, on the other hand, favor the use of desirability functions. Statistically significant lack of fit of some response functions indicates directions for further research related to using other empirical modeling tools, such as neural networks or kriging. However, this forces larger training data sets, which are difficult to obtain in industrial conditions.

The optimization algorithms recommended the following quasi-optimal combination of process parameters: 56.4% power, 3.1 mm distance of the hardening coil, and 793 mm/min of the feed rate when the industrial conditions were considered. Industrial conditions in the considered case forced the rejection of the possible global optimum of the desirability function, and the quasi-optimal solution was recommended, related to the local optimum, indicated approximately by both the classical optimization algorithm, such as SQP and the genetic algorithm. Uncertainties in empirical modeling also indicate that industrial conditions with limited experimental data (also with a lack of replication) make it difficult to determine optimal solutions. However, the confirmation experiments show that the quasi-optimal process is reliable and high-quality. The use of the nonparametric Wilcoxon test allowed the evaluation of a confirmatory experiment that provided a small sample of data, which is due to the industrial conditions of the research conducted. The test results confirm a statistically significant improvement in process quality in terms of the measures introduced.

The proposed hybrid approach in this paper can be considered an effective technique in some academic research and industrial applications to identify the optimal parameters in the induction hardening process and decrease production costs and time.

Industrial conditions should be carefully applied when solving optimization problems to meet multiple requirements. However, it can be concluded that the influence of treatment conditions on other responses of the technological process, such as total energy consumption, tool wear, residual stresses, etc., has not been investigated. Therefore, more goals should be considered during the formulation of optimization problems.

## Figures and Tables

**Figure 1 materials-16-05791-f001:**
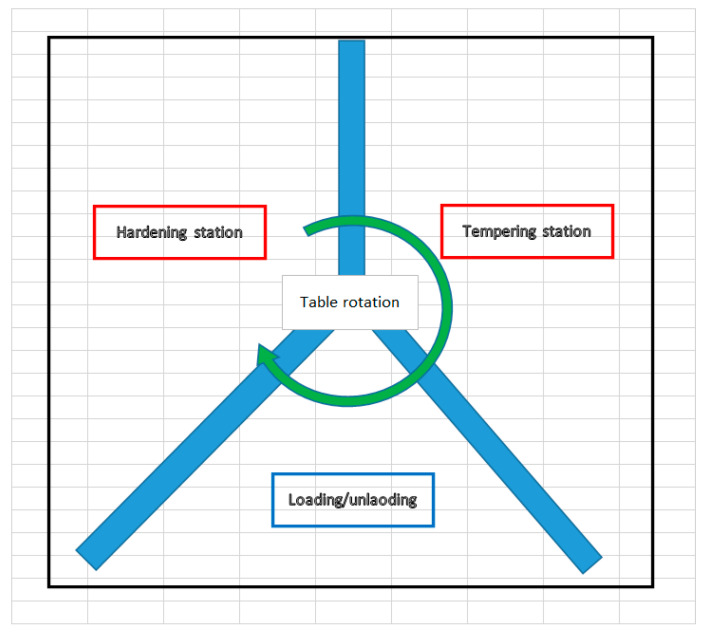
The automatic inductive hardening and tempering machine.

**Figure 2 materials-16-05791-f002:**
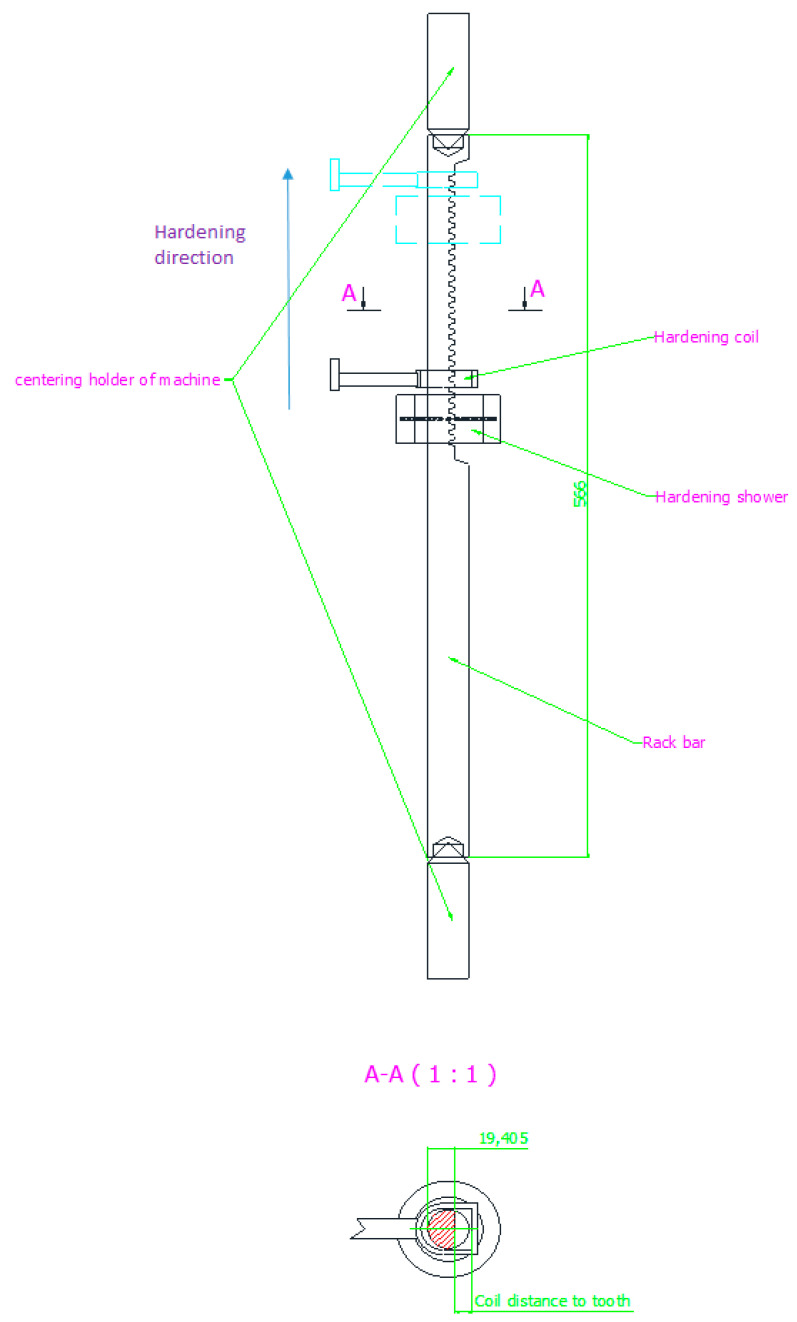
The process setup.

**Figure 3 materials-16-05791-f003:**
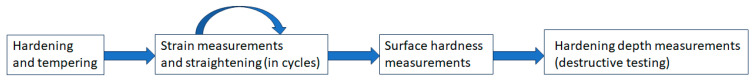
The general flow of the experimentation procedure.

**Figure 4 materials-16-05791-f004:**
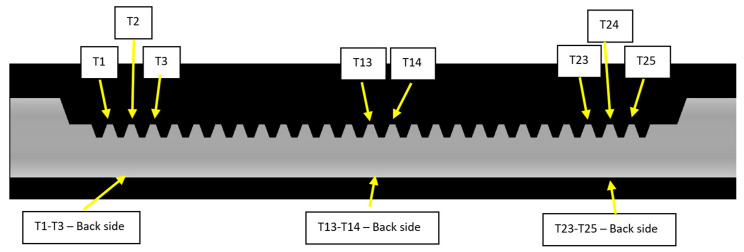
Zones for the hardness measurements of the rack bar after induction hardening operation.

**Figure 5 materials-16-05791-f005:**
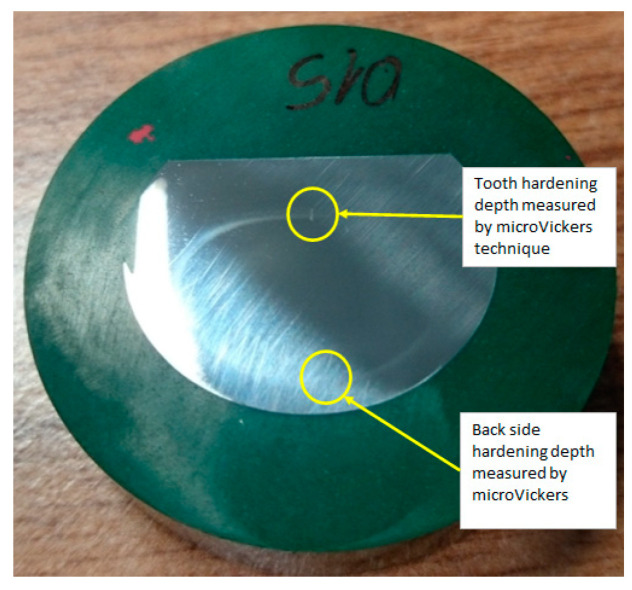
The setup of the hardness depth measurements.

**Figure 6 materials-16-05791-f006:**
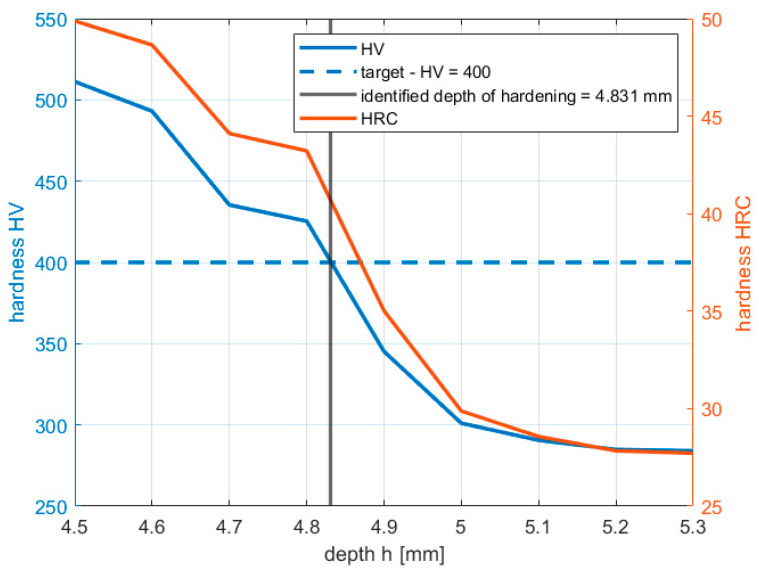
Example of hardness profile from the tooth side.

**Figure 7 materials-16-05791-f007:**
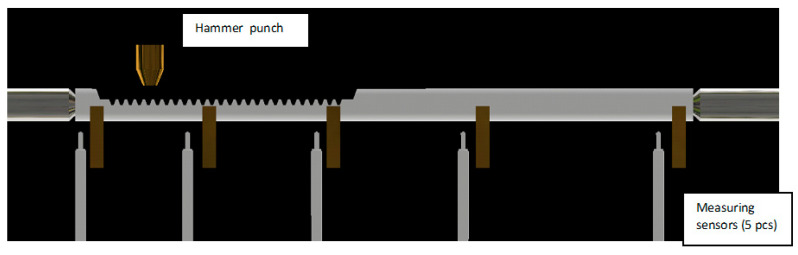
The setup of the deformation measurements.

**Figure 8 materials-16-05791-f008:**
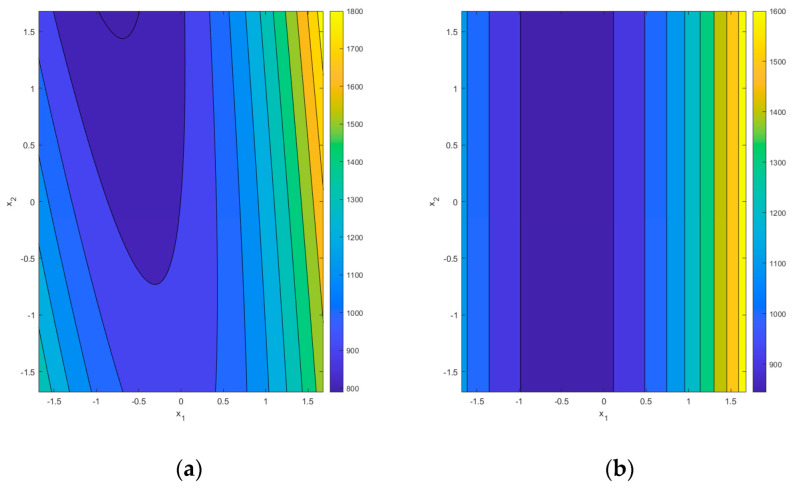
Contour plot for the slice *x*_3_ = 0.5 of the maximal deflection response hypersurface *y*_1_: (**a**) the full quadratic model; (**b**) the reduced model.

**Figure 9 materials-16-05791-f009:**
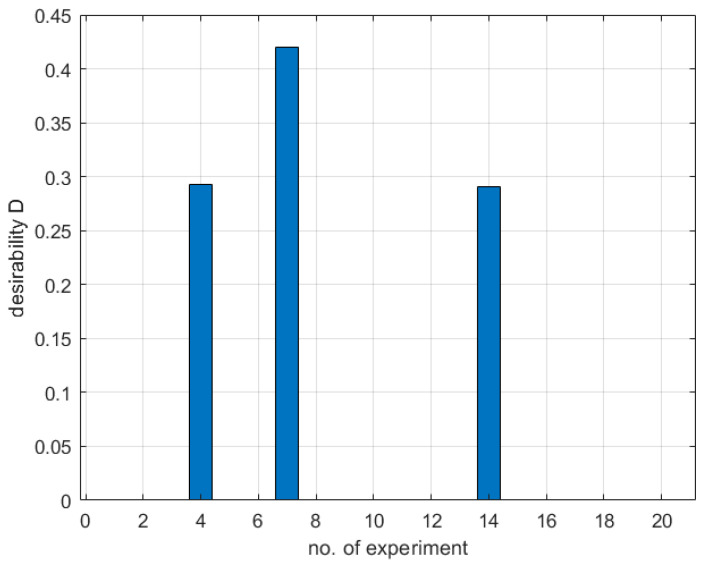
Desirability values for the experimental data obtained using CCD.

**Figure 10 materials-16-05791-f010:**
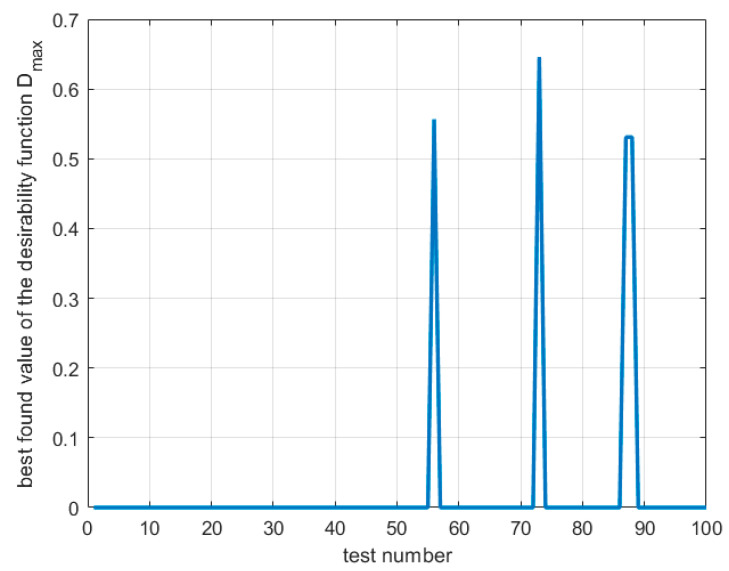
Best-found values of the desirability function for SQP optimization during N = 100 tests.

**Figure 11 materials-16-05791-f011:**
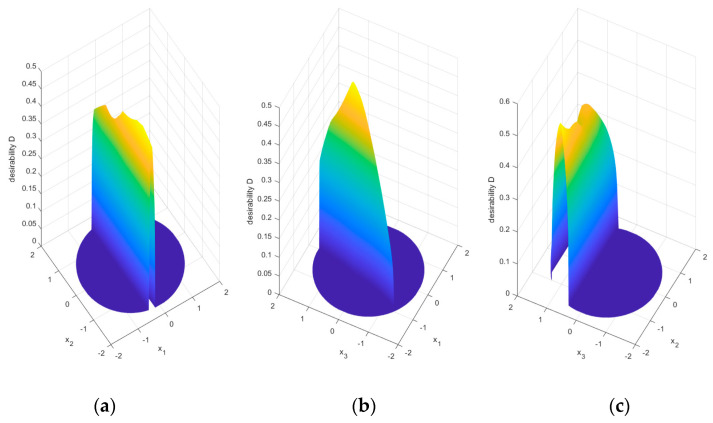
Slices of the desirability function hypersurface *D*(*x*_1_, *x*_2_, *x*_3_): (**a**) *x*_3_ = 0; (**b**) *x*_2_ = 0; (**c**) *x*_1_ = 0.

**Figure 12 materials-16-05791-f012:**
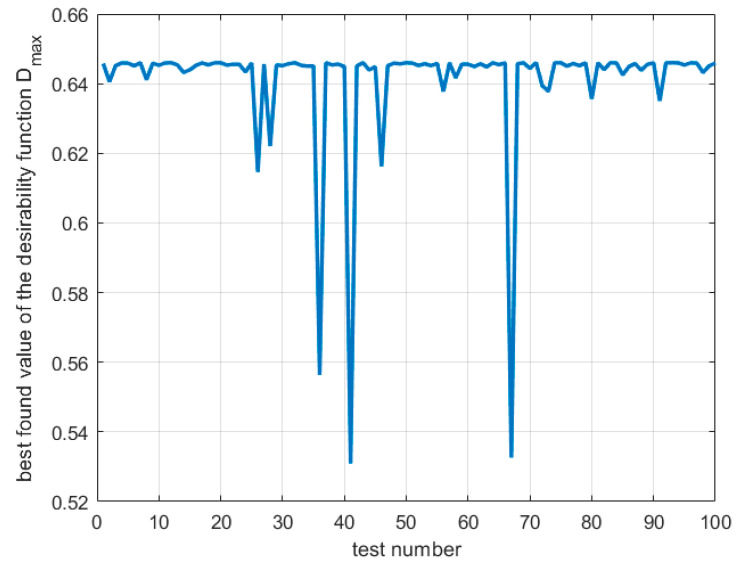
Best-found values of the desirability function for GA optimization during N = 100 tests.

**Figure 13 materials-16-05791-f013:**
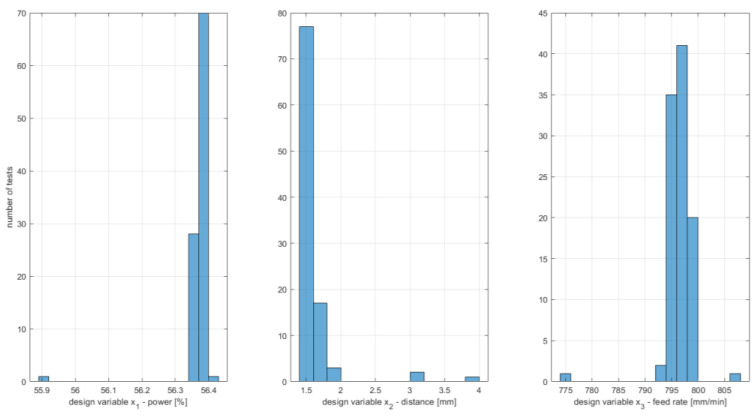
Histograms of optimal design variables for GA optimization results.

**Table 1 materials-16-05791-t001:** The CCD plan of experiments with process responses.

No.	*x* _1_	*x* _2_	*x* _3_	*y*_1_ [μm]	y_2_ [mm]	*y*_3_ [mm]	*y*_4_ [HRC]	*y*_5_ [HRC]	*y*_6_ [HRC]	*y*_7_ [HRC]	*y*_8_ [HRC]	*y*_9_ [HRC]
1	1	−1	−1	1721	10.358	5.098	47.9	50.0	53.3	47.4	52.7	54.5
2	−1	1	−1	720	4.008	1.416	56.1	57.6	58.0	55.0	55.1	55.6
3	1	1	1	1258	4.831	2.420	54.0	57.5	58.0	53.7	54.8	56.0
4	0	0	0	962	4.810	1.954	55.2	57.2	58.2	54.4	54.5	54.5
5	−1	−1	1	958	3.776	0.351	56.4	58.8	58.8	40.4	40.0	40.9
6	0	0	0	1160	4.767	1.957	56.0	58.6	58.9	54.0	54.9	55.9
7	−1	−1	−1	948	4.784	1.451	55.3	57.8	58.8	53.5	54.5	53.8
8	−1	1	1	1213	4.771	1.987	56.5	59.6	59.8	39.4	37.7	39.1
9	0	0	0	987	2.933	0.234	55.1	57.8	58.7	52.5	54.7	55.2
10	1	1	−1	2005	7.250	4.507	52.2	56.1	56.7	50.8	54.3	55.8
11	1	−1	1	981	5.778	2.512	53.4	56.5	57.1	52.2	55.0	54.7
12	0	0	0	1020	4.672	1.946	55.1	57.0	58.3	54.2	54.6	55.5
13	1.682	0	0	2131	7.704	4.579	46.2	51.2	56.1	46.3	51.6	51.5
14	0	0	1.682	745	4.059	1.174	56.4	58.3	58.9	54.6	54.3	54.5
15	0	0	0	1085	4.818	1.932	56.4	58.5	59.0	52.3	55.3	56.3
16	−1.682	0	0	1089	2.953	0.000	56.9	58.5	58.8	27.7	27.4	26.5
17	0	0	0	1013	4.766	1.969	55.6	57.4	58.0	54.0	55.2	56.1
18	0	1.682	0	821	4.331	1.938	55.7	58.2	58.6	55.1	56.0	56.0
19	0	−1.682	0	1435	5.669	1.991	55.0	57.3	57.8	53.0	55.1	56.0
20	0	0	−1.682	2057	6.132	3.267	52.1	55.4	56.0	52.9	55.0	56.4

**Table 2 materials-16-05791-t002:** Analysis of variance for the mathematical model of maximal deflection response *y*_1_.

Source	Sum of Squares	Sum of Squares Contribution [%]	*df*	Mean Square	*F*-Value	*p*-Value	Notes
Model	2,908,285.706	82.568	9	323,142.856	5.263	0.008	S
*x* _1_	1,101,450.148	31.271	1	1,101,450.148	17.939	0.002	S
*x* _2_	14,482.149	0.411	1	14,482.149	0.236	0.638	
*x* _3_	745,417.899	21.163	1	745,417.899	12.140	0.006	S
*x* _1_ *x* _2_	35,644.500	1.012	1	35,644.500	0.581	0.464	
*x* _1_ *x* _3_	495,012.500	14.054	1	495,012.500	8.062	0.018	S
*x* _2_ *x* _3_	28,322.000	0.804	1	28,322.000	0.461	0.512	
*x* _1_ ^2^	394,238.133	11.193	1	394,238.133	6.421	0.030	S
*x* _2_ ^2^	361.006	0.010	1	361.006	0.006	0.940	
*x* _3_ ^2^	120,678.769	3.426	1	120,678.769	1.965	0.191	
Residual	613,997.244	17.432	10	61,399.724			
Lack of fit	587,578.410	16.682	5	117,515.682	22.241	0.002	S
Pure error	26,418.833	0.750	5	5283.767			
Total	3,522,282.950	100.000	19	185,383.313			

**Table 3 materials-16-05791-t003:** Analysis of variance for the mathematical model of the hardening depth of teeth *y*_2_.

Source	Sum of Squares	Sum of Squares Contribution [%]	*df*	Mean Square	*F*-Value	*p*-Value	Notes
Model	51.625	92.841	9	5.736	14.410	<0.001	S
*x* _1_	26.068	46.881	1	26.068	65.488	<0.001	S
*x* _2_	2.712	4.878	1	2.712	6.814	0.026	S
*x* _3_	8.431	15.162	1	8.431	21.180	0.001	S
*x* _1_ *x* _2_	2.283	4.106	1	2.283	5.736	0.038	S
*x* _1_ *x* _3_	5.702	10.254	1	5.702	14.325	0.004	S
*x* _2_ *x* _3_	1.933	3.476	1	1.933	4.855	0.052	
*x* _1_ ^2^	2.479	4.458	1	2.479	6.227	0.032	S
*x* _2_ ^2^	1.285	2.311	1	1.285	3.228	0.103	
*x* _3_ ^2^	1.592	2.863	1	1.592	3.999	0.073	
Residual	3.981	7.159	10	0.398			
Lack of fit	1.165	2.096	5	0.233	0.414	0.822	
Pure error	2.815	5.063	5	0.563			
Total	55.606	100.000	19	2.927			

**Table 4 materials-16-05791-t004:** Analysis of variance for the mathematical model of the hardening depth on the back side *y*_3_.

Source	Sum of Squares	Sum of Squares Contribution [%]	*df*	Mean Square	*F*-Value	*p*-Value	Notes
Model	32.331	91.339	9	3.592	11.718	<0.001	S
*x* _1_	21.244	60.016	1	21.244	69.295	<0.001	S
*x* _2_	0.050	0.142	1	0.050	0.164	0.694	
*x* _3_	5.570	15.737	1	5.570	18.170	0.002	S
*x* _1_ *x* _2_	0.652	1.842	1	0.652	2.127	0.175	
*x* _1_ *x* _3_	2.147	6.064	1	2.147	7.002	0.024	S
*x* _2_ *x* _3_	0.589	1.663	1	0.589	1.920	0.196	
*x* _1_ ^2^	1.127	3.185	1	1.127	3.677	0.084	
*x* _2_ ^2^	0.391	1.106	1	0.391	1.277	0.285	
*x* _3_ ^2^	0.939	2.654	1	0.939	3.064	0.111	
Residual	3.066	8.661	10	0.307			
Lack of fit	0.606	1.713	5	0.121	0.247	0.925	
Pure error	2.459	6.948	5	0.492			
Total	35.397	100.000	19	1.863			

**Table 5 materials-16-05791-t005:** Analysis of variance for the mathematical model of the hardness of the teeth in the first zone *y*_4_.

Source	Sum of Squares	Sum of Squares Contribution (%)	*df*	Mean Square	*F*-Value	*p*-Value	Notes
Model	146.920	94.911	9	16.324	20.722	<0.001	S
*x* _1_	88.654	57.271	1	88.654	112.539	<0.001	S
*x* _2_	3.564	2.303	1	3.564	4.525	0.059	
*x* _3_	18.820	12.158	1	18.820	23.890	0.001	S
*x* _1_ *x* _2_	2.000	1.292	1	2.000	2.539	0.142	
*x* _1_ *x* _3_	4.205	2.716	1	4.205	5.338	0.043	S
*x* _2_ *x* _3_	2.420	1.563	1	2.420	3.072	0.110	
*x* _1_ ^2^	25.952	16.765	1	25.952	32.944	<0.001	S
*x* _2_ ^2^	0.000	0.000	1	0.000	0.000	0.995	
*x* _3_ ^2^	2.163	1.397	1	2.163	2.746	0.129	
Residual	7.878	5.089	10	0.788			
Lack of fit	6.424	4.150	5	1.285	4.420	0.064	
Pure error	1.453	0.939	5	0.291			
Total	154.798	100.000	19	8.147			

**Table 6 materials-16-05791-t006:** Analysis of variance for the mathematical model of the hardness of the teeth in the second zone *y*_5_.

Source	Sum of Squares	Sum of Squares Contribution (%)	*df*	Mean Square	*F*-Value	*p*-Value	Notes
Model	98.206	90.424	9	10.912	10.492	<0.001	S
*x* _1_	49.412	45.497	1	49.412	47.512	<0.001	S
*x* _2_	6.216	5.723	1	6.216	5.977	0.035	S
*x* _3_	18.226	16.782	1	18.226	17.525	0.002	S
*x* _1_ *x* _2_	5.281	4.863	1	5.281	5.078	0.048	S
*x* _1_ *x* _3_	3.001	2.763	1	3.001	2.886	0.120	
*x* _2_ *x* _3_	2.101	1.935	1	2.101	2.020	0.186	
*x* _1_ ^2^	13.157	12.114	1	13.157	12.651	0.005	S
*x* _2_ ^2^	0.070	0.065	1	0.070	0.067	0.800	
*x* _3_ ^2^	0.889	0.819	1	0.889	0.855	0.377	
Residual	10.400	9.576	10	1.040			
Lack of fit	8.125	7.481	5	1.625	3.571	0.094	
Pure error	2.275	2.095	5	0.455			
Total	108.606	100.000	19	5.716			

**Table 7 materials-16-05791-t007:** Analysis of variance for the mathematical model of the hardness of the teeth in the third zone *y*_6_.

Source	Sum of Squares	Sum of Squares Contribution (%)	*df*	Mean Square	*F*-Value	*p*-Value	Notes
Model	36.465	89.467	9	4.052	9.438	<0.001	S
*x* _1_	16.127	39.568	1	16.127	37.566	<0.001	S
*x* _2_	2.502	6.138	1	2.502	5.828	0.036	S
*x* _3_	10.156	24.918	1	10.156	23.658	0.001	S
*x* _1_ *x* _2_	2.101	5.155	1	2.101	4.895	0.051	
*x* _1_ *x* _3_	1.361	3.340	1	1.361	3.171	0.105	
*x* _2_ *x* _3_	0.061	0.150	1	0.061	0.143	0.714	
*x* _1_ ^2^	2.257	5.536	1	2.257	5.256	0.045	S
*x* _2_ ^2^	0.246	0.603	1	0.246	0.572	0.467	
*x* _3_ ^2^	2.257	5.536	1	2.257	5.256	0.045	S
Residual	4.293	10.533	10	0.429			
Lack of fit	3.465	8.501	5	0.693	4.183	0.071	
Pure error	0.828	2.032	5	0.166			
Total	40.758	100.000	19	2.145			

**Table 8 materials-16-05791-t008:** Analysis of variance for the mathematical model of the hardness of the back-side in the first zone *y*_7_.

Source	Sum of Squares	Sum of Squares Contribution (%)	*df*	Mean Square	*F*-Value	*p*-Value	Notes
Model	840.725	91.064	9	93.414	11.323	<0.001	S
*x* _1_	162.321	17.582	1	162.321	19.676	0.001	S
*x* _2_	5.841	0.633	1	5.841	0.708	0.420	
*x* _3_	24.094	2.610	1	24.094	2.921	0.118	
*x* _1_ *x* _2_	2.420	0.262	1	2.420	0.293	0.600	
*x* _1_ *x* _3_	165.620	17.939	1	165.620	20.076	0.001	S
*x* _2_ *x* _3_	2.420	0.262	1	2.420	0.293	0.600	
*x* _1_ ^2^	455.825	49.373	1	455.825	55.254	<0.001	S
*x* _2_ ^2^	2.348	0.254	1	2.348	0.285	0.605	
*x* _3_ ^2^	1.276	0.138	1	1.276	0.155	0.702	
Residual	82.497	8.936	10	8.250			
Lack of fit	78.284	8.479	5	15.657	18.580	0.003	S
Pure error	4.213	0.456	5	0.843			
Total	923.222	100.000	19	48.591			

**Table 9 materials-16-05791-t009:** Analysis of variance for the mathematical model of the hardness of the back-side in the second zone *y*_8_.

Source	Sum of Squares	Sum of Squares Contribution (%)	*df*	Mean Square	*F*-Value	*p*-Value	Notes
Model	995.989	92.275	9	110.665	13.273	<0.001	S
*x* _1_	360.856	33.432	1	360.856	43.280	<0.001	S
*x* _2_	0.108	0.010	1	0.108	0.013	0.912	
*x* _3_	67.118	6.218	1	67.118	8.050	0.018	S
*x* _1_ *x* _2_	1.201	0.111	1	1.201	0.144	0.712	
*x* _1_ *x* _3_	150.511	13.944	1	150.511	18.052	0.002	S
*x* _2_ *x* _3_	2.761	0.256	1	2.761	0.331	0.578	
*x* _1_ ^2^	395.712	36.662	1	395.712	47.461	<0.001	S
*x* _2_ ^2^	2.716	0.252	1	2.716	0.326	0.581	
*x* _3_ ^2^	0.194	0.018	1	0.194	0.023	0.882	
Residual	83.376	7.725	10	8.338			
Lack of fit	82.843	7.675	5	16.569	155.330	<0.001	S
Pure error	0.533	0.049	5	0.107			
Total	1079.366	100.000	19	56.809			

**Table 10 materials-16-05791-t010:** Analysis of variance for the mathematical model of the hardness of the back-side in the third zone *y*_9_.

Source	Sum of Squares	Sum of Squares Contribution (%)	*df*	Mean Square	*F*-Value	*p*-Value	Notes
Model	1053.639	92.749	9	117.071	14.213	<0.001	S
*x* _1_	397.146	34.960	1	397.146	48.215	<0.001	S
*x* _2_	0.495	0.044	1	0.495	0.060	0.811	
*x* _3_	75.893	6.681	1	75.893	9.214	0.013	S
*x* _1_ *x* _2_	0.845	0.074	1	0.845	0.103	0.755	
*x* _1_ *x* _3_	111.005	9.771	1	111.005	13.477	0.004	S
*x* _2_ *x* _3_	1.620	0.143	1	1.620	0.197	0.667	
*x* _1_ ^2^	444.042	39.088	1	444.042	53.909	<0.001	S
*x* _2_ ^2^	3.038	0.267	1	3.038	0.369	0.557	
*x* _3_ ^2^	1.010	0.089	1	1.010	0.123	0.733	
Residual	82.369	7.251	10	8.237			
Lack of fit	80.161	7.056	5	16.032	36.299	0.001	S
Pure error	2.208	0.194	5	0.442			
Total	1136.008	100.000	19	59.790			

**Table 11 materials-16-05791-t011:** Statistical characteristics of regression models for responses of the process.

No.	Model	Statistics				
		Multiple R	R^2^	Adjusted R^2^	*p*-Value	PRESS
1	The maximal deflection, *y*_1_	0.909	0.826	0.669	0.008	4,540,000
2	Hardening depth of teeth, *y*_2_	0.964	0.928	0.864	<0.001	12.891
3	The hardening depth on the back side, *y*_3_	0.956	0.913	0.836	<0.001	8.402
4	The hardness of the teeth in the first zone, *y*_4_	0.956	0.913	0.836	<0.001	52.643
5	The hardness of the teeth in the second zone, *y*_5_	0.950	0.902	0.813	<0.001	72.291
6	The hardness of the teeth in the third zone, *y*_6_	0.948	0.898	0.807	<0.001	31.142
7	The hardness of the back-side in the first zone, *y*_7_	0.954	0.911	0.830	<0.001	600.672
8	The hardness of the back-side in the first zone, *y*_8_	0.961	0.923	0.853	<0.001	630.241
9	The hardness of the back-side in the first zone, *y*_9_	0.963	0.928	0.862	<0.001	614.364

**Table 12 materials-16-05791-t012:** The reduced models of responses.

No.	Model	Equation	R^2^	PRESS
1	The maximal deflection, *y*_1_	$y_{1}=1107.86+283.98x_{1}-233.62x_{3}+157.55x_{1}^{2}-248.75x_{1}x_{3}$	0.771	1,520,000
2	Hardening depth of teeth, *y*_2_	$\begin{array}{l} y_{2}=4.45+1.38x_{1}-0.45x_{2}-0.79x_{3}+0.42x_{1}^{2}+0.30x_{2}^{2}+0.33x_{3}^{2}\\ -0.53x_{1}x_{2}-0.84x_{1}x_{3}+0.49x_{2}x_{3} \end{array}$	0.928	12.891
3	The hardening depth on the back side, *y*_3_	$\begin{array}{l} y_{3}=1.66+1.25x_{1}+0.06x_{2}-0.64x_{3}+0.28x_{1}^{2}+0.17x_{2}^{2}+0.26x_{3}^{2}\\ -0.29x_{1}x_{2}-0.52x_{1}x_{3}+0.27x_{2}x_{3} \end{array}$	0.913	8.402
4	The hardness of the teeth in the first zone, *y*_4_	$y_{4}=55.27-2.55x_{1}+1.18x_{3}-1.31x_{1}^{2}$	0.863	43.793
5	The hardness of the teeth in the second zone, *y*_5_	$y_{5}=57.59-1.89x_{1}+0.68x_{2}+1.16x_{3}-0.94x_{1}^{2}$	0.801	41.591
6	The hardness of the teeth in the third zone, *y*_6_	$y_{6}=58.41-1.09x_{1}+0.43x_{2}+0.85x_{3}-0.38x_{1}^{2}-0.38x_{3}^{2}+0.52x_{1}x_{2}$	0.862	17.054
7	The hardness of the back-side in the first zone, *y*_7_	$y_{7}=54.05+3.45x_{1}-5.69x_{1}^{2}+4.55x_{1}x_{3}$	0.874	307.236
8	The hardness of the back-side in the second zone, *y*_8_	$y_{8}=55.25+5.14x_{1}-2.22x_{3}-5.29x_{1}^{2}+4.34x_{1}x_{3}$	0.921	273.008
9	The hardness of the back-side in the third zone, *y*_9_	$y_{9}=56.08+5.39x_{1}-2.36x_{3}-5.62x_{1}^{2}+3.725x_{1}x_{3}$	0.922	273.822

**Table 13 materials-16-05791-t013:** Parameters of the desirability functions for the identified response models.

No.	$\mathrm{Response}\;{\hat{y}}_{i}$	Type	*L_i_*	*T_i_*	*U_i_*	*s*	*w_i_*
1.	*f* _max_	STB	1.0 mm	-	1.4 mm	2	1
2.	*h_t_*	NTB	3.9 mm	4.1 mm	10.0 mm	2	0.8
3.	*h_b_*	NTB	1.0 mm	1.4 mm	10.0 mm	2	0.8
4.	HRC_t1_	NTB	55 HRC	57.5 HRC	60 HRC	2	0.6
5.	HRC_t2_	NTB	55 HRC	57.5 HRC	60 HRC	2	0.6
6.	HRC_t3_	NTB	55 HRC	57.5 HRC	60 HRC	2	0.6
7.	HRC_b1_	NTB	52 HRC	53.5 HRC	55 HRC	2	0.6
8.	HRC_b2_	NTB	52 HRC	53.5 HRC	55 HRC	2	0.6
9.	HRC_b3_	NTB	52 HRC	53.5 HRC	55 HRC	2	0.6

**Table 14 materials-16-05791-t014:** Statistics of the SQP optimization results for the desirability function values.

Statistics	Value
Sample mean value, $\bar{x}$	0.023
Sample standard deviation, σ	0.112
Sample median, *M*	0.000
Sample minimum, min(max(*D*))	0.000
Sample maximum, max(max(*D*))	0.646

**Table 15 materials-16-05791-t015:** Statistics of the evolutionary optimization results for the desirability function values.

Statistics	Value
Sample mean value, $\bar{x}$	0.641
Sample standard deviation, σ	0.019
Sample median, *M*	0.646
Sample minimum, min(max(*D*))	0.531
Sample maximum, max(max(*D*))	0.646

**Table 16 materials-16-05791-t016:** Results of the confirmation experiment.

No.	*y*_1_ [μm]	*y*_2_ [mm]	*y*_3_ [mm]	*y*_4_ [HRC]	*y*_5_ [HRC]	*y*_6_ [HRC]	*y*_7_ [HRC]	*y*_8_ [HRC]	*y*_9_ [HRC]	*D*
1	605	4.190	2.010	57.3	57.1	57.5	54.3	54.1	54.4	0.606
2	815	4.522	2.063	56.9	57.1	57.5	53.4	55.3	56.2	0.000
3	773	4.410	2.420	57.8	56.5	57.4	54.3	54.6	54.1	0.508
4	681	4.544	2.092	56.8	56.8	57.1	53.3	54.9	55.5	0.000
5	656	4.368	1.904	56.6	56.7	56.9	53.4	54.7	54.1	0.514
6	635	4.590	2.070	58.1	57.3	58.1	53.8	54.2	52.6	0.605

**Table 17 materials-16-05791-t017:** Results of the Wilcoxon test for data from the confirmatory experiment.

No.	Hypothesis *H*_1_	*p*-Value	Result
1	*M*(**D**^conf^) ≠ max(*D*_opt_)	0.031	the data in the sample come from a continuous distribution with a median different than the best optimal solution of 0.646
2	*M*((**y**_1_)^conf^) < *L*(*y*_1_)	0.016	the data in the sample come from a continuous distribution with a median less than the lower bound for the maximal deflection response of 1000 μm
3	*M*((**y**_2_)^conf^) > *L*(*y*_2_)	0.016	the data in the sample come from a continuous distribution with a median greater than the lower bound for the hardening depth of teeth of 3.9 mm
4	*M*((**y**_3_)^conf^) > *L*(*y*_3_)	0.016	the data in the sample come from a continuous distribution with a median greater than the lower bound for the hardening depth of the back side of 1.0 mm
5	*M*([**y**_4_; **y**_5_; **y**_6_])^conf^) > *L*(*y*_4_)	<0.001	the data in the sample come from a continuous distribution with a median greater than the lower bound for the hardness of the teeth in I, II and III zones of 55 HRC
6	*M*([**y**_4_; **y**_5_; **y**_6_])^conf^) < *U*(*y*_4_)	<0.001	the data in the sample come from a continuous distribution with a median less than the upper bound for the hardness of the teeth in the I, II and III zones of 60 HRC
7	*M*([**y**_7_; **y**_8_; **y**_9_])^conf^) > *L*(*y*_7_)	<0.001	the data in the sample come from a continuous distribution with a median greater than the lower bound for the hardness of the back side in the I, II and III zones of 52 HRC
8	*M*([**y**_7_; **y**_8_; **y**_9_])^conf^) < *U*(*y*_7_)	0.003	the data in the sample come from a continuous distribution with a median less than the upper bound for the hardness of the back side in the I, II and III zones of 55 HRC

## Data Availability

All data generated or analyzed in this research were included in this published article.

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
