# Peer review of "Evolutionary Optimizing Process Parameters in the Induction Hardening of Rack Bar by Response Surface Methodology and Desirability Function Approach under Industrial Conditions"

_materials, 2023, doi:10.3390/ma16175791_

Round 1
Reviewer 1 Report
An evolutionary strategy in the form of a floating-point encoded genetic algorithm was used, which exhibits a non-zero probability of obtaining a global extremum and is a gradient-free method. Confirmation experiments show the improvement of the process quality using introduced measures. Some comments given to the authors as follows:
1. Line 30, The novelty in the current article by the authors is too weak. The past has seen extensive published work of written material. It is required to provide more details for more explanation about the present novel in the introductory section.
2. Line 42-132, this paragraph is too long and monotonous with not easy captured the main information. It is suggested to split into several paragraph with more focus discussion in each paragraph.
3. Line 49-51, as the authors explaining regarding computational simulation, please explain the urgency of this approach. It brings several advantages such as lower cost and faster results camped to experimental and analytical approach. Provide this information alowng with relevant reference as follows: https://doi.org/10.3390/biomedicines11030951 and https://doi.org/10.1080/23311916.2023.2218691
4. Line 171, where is discussion for the automatic inductive hardening and tempering machine? Please provide it for better understanding.
5. Line 174-176, what is the basis for this adjustment? It is initiated by authors? Or referred from previous literature?
6. Line 201, detail engineering drawing with detail dimension is needed for reproducibility.
7. Since the present study adopted response surface methodology for their investigation, it is worth to mention recently published work using similar method to explain state of the art in the present study, please incorporate this one: https://doi.org/10.3390/ma16124458
-
Reviewer 2 Report
-Introduction part is big. Shorten it by providing relevant information only.
-Literature survey should be provided with the methodologies and results adopted by the previous researchers. A lot of work has been done on induction hardening operations.
- Literature gap is missing. State them clearly.
-Why RSM chosen for this work? Why has CCD been chosen over BBD?
-Add a flowchart that depicts your experimental methodology.
-Contour plot should be clear with the axis. Relavent zones should be discussed.
-Why you chosen DFA? There are many other statistical methods. State the significance.
-GA fitness plot is not converging. You should have chosen more generations and tested for a larger population. The plot should become constant after some point, which can help in achieving fitness value.
-Add ANOVA results and discuss the significant factors and their influence.
-How your results comparable? Provide a convergence of your results with previous research.
-Add some future scope. How your studies can be expanded
-Add the applicability of your research.
Round 2
Reviewer 2 Report
-Change the ANOVA values up to 3 decimal places. Write in the standard form, e.g., 1.111.
-Add the % contribution column to the ANOVA table.
-You can add Table 17 at the end of the results and discussion sections and write a brief note about it. In coclusion, it can be added with fewer significant results only.
